Health-related quality of life and associated factors after hip fracture. Results from a six-month prospective cohort study

Deutschbein Johannes johannes.deutschbein@charite.de 1
Lindner Tobias 2
Möckel Martin 2
Pigorsch Mareen 3
Gilles Gabriela 1
Stöckle Ulrich 4
Müller-Werdan Ursula 5
Schenk Liane 1
1 Charité—Universitätsmedizin Berlin, corporate member of Freie Universität Berlin and Humboldt-Universität zu Berlin, Institute of Medical Sociology and Rehabilitation Science , Germany
2 Charité—Universitätsmedizin Berlin, corporate member of Freie Universität Berlin and Humboldt-Universität zu Berlin, Division of Emergency Medicine Campus Mitte and Virchow , Germany
3 Charité—Universitätsmedizin Berlin, corporate member of Freie Universität Berlin and Humboldt-Universität zu Berlin, Institute of Biometry and Clinical Epidemiology , Germany
4 Charité—Universitätsmedizin Berlin, corporate member of Freie Universität Berlin and Humboldt-Universität zu Berlin, Center for Musculosceletal Surgery (CMSC) , Germany
5 Charité—Universitätsmedizin Berlin, corporate member of Freie Universität Berlin and Humboldt-Universität zu Berlin, Department of Geriatrics and Medical Gerontology , Germany ,
Mallhi Tauqeer
Electronic publication date: 2023 Mar 15
Publication date: 2023
Volume: 11
Electronic Location ID: e14671
Received 2022 Jun 24; Accepted 2022 Dec 11
Copyright: 2023 Deutschbein et al.
Copyright year: 2023
Copyright holder: Deutschbein et al.
License: This is an open access article distributed under the terms of the Creative Commons Attribution License, which permits unrestricted use, distribution, reproduction and adaptation in any medium and for any purpose provided that it is properly attributed. For attribution, the original author(s), title, publication source (PeerJ) and either DOI or URL of the article must be cited.
License URL: https://creativecommons.org/licenses/by/4.0/

Keywords: Health-related quality of life, Hip fracture, Elderly, Health services research, Health care, Patient-reported outcomes, Quality of life, Observational study

Funding: The Federal Ministry of Education and Research (BMBF) 01GY1604 Open Access Publication Fund of Charité—Universitätsmedizin Berlin The German Research Foundation (DFG) This study was funded by the Federal Ministry of Education and Research (BMBF), grant number 01GY1604. We received financial support from the Open Access Publication Fund of Charité—Universitätsmedizin Berlin and the German Research Foundation (DFG). The funders had no role in study design, data collection and analysis, decision to publish, or preparation of the manuscript.

==============================
Background

Hip fractures are a major public health problem with increasing relevance in aging societies. They are associated with high mortality rates, morbidity, and loss of independence. The aim of the EMAAge study was to determine the impact of hip fractures on patient-reported health-related quality of life (HRQOL), and to identify potential risk factors for worse outcomes.

Methods

EMAAge is a multicenter, prospective cohort study of patients who suffered a hip fracture. Patients or, if necessary, proxies were interviewed after initial treatment and after six months using standardized questionnaires including the EQ-5D-5L instrument, the Oxford Hip Score, the PHQ-4, the Short Nutritional Assessment Questionnaire, and items on patients living situation. Medical data on diagnoses, comorbidities, medications, and hospital care were derived from hospital information systems.

Results

A total of 326 patients were included. EQ-5D index values decreased from a mean of 0.70 at baseline to 0.63 at six months. The mean self-rated health on the EQ-VAS decreased from 69.9 to 59.4. Multivariable linear regression models revealed three relevant associated factors with the six-months EQ-5D index: symptoms of depression and anxiety, pre-fracture limitations in activities of daily living, and no referral to a rehabilitation facility had a negative impact. In addition, the six-months EQ-VAS was negatively associated with polypharmacy, living in a facility, and migration background.

Conclusions

Hip fractures have a substantial negative impact on patients HRQOL. Our results suggest that there are modifying factors that need further investigation including polypharmacy and migration background. Structured and timely rehabilitation seems to be a protective factor.

Introduction

Hip fractures are serious injuries requiring emergency care, mostly surgery, and complex rehabilitation efforts. They can occur at any age but are primarily affecting older people with osteoporosis. Typically, they are associated with deterioration of strength and equilibrium, and caused by falls (Terroso et al., 2014). Among osteoporotic fractures, hip fractures are the most serious with 1-year-mortality rates between 10% and 20% (Abrahamsen et al., 2009), and enormous societal costs (Cummings & Melton, 2002; Haentjens, Lamraski & Boonen, 2005). Estimates assume that worldwide around 18% of women and 6% of men are affected by hip fractures during their lifetime. Because women are affected more often by risk factors such as osteoporosis and have a higher life expectancy, about 75% of hip fractures are experienced by women (Cummings & Melton, 2002). With increasing life expectancy, case numbers are projected to mount up to 4.5 million cases in 2050 (Veronese & Maggi, 2018). Thus, hip fractures are one of the major public health problems of ageing societies.

For a large share of patients, hip fractures have dramatic consequences. Only 40 to 60% recover their pre-fracture level of mobility, 20 to 60% lose their independence in self-care, and up to 20% need to be institutionalized (Dyer et al., 2016; Knauf et al., 2019). In terms of disability-adjusted life years (DALYs), this equates a loss of 27 DALYs per 1,000 patients due to hip fractures, which is similar to diseases like breast and pancreatic cancer (Papadimitriou et al., 2017).

Increasingly, hip fracture research has focused on patient-centered outcomes such as health-related quality of life. It has been stated that the true burden of these fractures is underestimated without the considering HRQOL, and that there is a need for studies investigating this outcome (Xenodemetropoulos et al., 2004). So far, many studies have confirmed that hip fractures have a strong and persistent negative impact on patients’ HRQOL (Campenfeldt et al., 2020; Peeters et al., 2016). However, only a few studies have been conducted within the German health care context (Buecking et al., 2014; Hack et al., 2019). In addition, little is known about the specific role of associated factors that modify the enduring negative impact of hip fractures on HRQOL. In particular, social factors and typical geriatric syndromes such as malnutrition, mental health problems, and polypharmacy are often neglected.

The aim of the EMAAge study was to reach a better understanding of the burden of hip fractures regarding HRQOL in Germany. We sought to identify potential risk factors for worse outcomes to help to improve patient-centered health care for hip fracture patients.

Materials & Methods

Study design

EMAAge is a multicenter, prospective, observational cohort study focusing on emergency patients with hip fractures. It is based on standardized interviews with patients or proxies at baseline and a follow-up interview supplemented by clinical data on diagnoses and treatment. The observational period was six months.

Due to the observatory and exploratory character of the study, the sample size was calculated based on feasibility considerations and a generalized power calculation. Given the average number of hip fracture cases in the recruitment area, the target sample size was set at 350.

The study is part of the research network EMANET—Emergency and Acute Medicine Network for Health Care Research, which is funded by the Federal Ministry of Education and Research (BMBF). It involves the eight Emergency Departments (ED) in the central district of Berlin, Germany (Berlin-Mitte) (Schmiedhofer et al., 2018). Two EDs are part of a university hospital, six of general hospitals.

The study protocol was registered in the German Clinical Trials Register (DRKS00014273). The ethics committee of Charité—Universitätsmedizin Berlin (EA1/362/16) approved of this study.

Setting and participants

Participants were consecutively recruited between June 2017 and June 2019 in six of eight hospitals of the network which provide orthopedic & trauma surgery. All patients admitted to one of the ED with “hip or pelvic pain” and “fall” were identified and assessed for inclusion. Patients were eligible for enrollment if (1) 18 years or older and (2) having an ED diagnosis of hip fracture based on the International Classification of Diseases, 10th revision (ICD-10): S72.0, S72.1 and S72.2.

The exclusion criteria were (1) refusal of the patient or their proxy to participate, (2) life-threating conditions during hospitalization, (3) limited proficiency in one of the questionnaire languages (German, English, Arabic, and Turkish).

Perioperative care was provided in general EDs, Intensive Care Units (ICUs) and orthopedic & trauma wards without special focus on geriatric care.

Patient recruitment

Eligible patients were approached by trained study nurses during the course of their in-hospital stay and after initial treatment of the fracture. After education and given written consent, study nurses carried out a standardized, tablet-based, face-to-face bedside interview with participating patients. Responses were entered by the interviewer and were directly transferred to a secure database server. When patients were unable to consent to participate and to provide reliable information due to cognitive impairment or dementia, a proxy was asked to give consent and information on behalf of the patient. This was either a close relative or a legal guardian. The latter were provided with a short version of the questionnaire excluding those items only a person itself or a close relative could assess. Proxy versions of survey instruments were used if available.

Clinical data on ED and in-hospital care and diagnoses from hospital information systems were assessed. Hospital data were systematically collected after discharge and entered into a standardized electronic clinical report form (eCRF). Manual double data entry of clinical data was performed to ensure reliability and accuracy. The follow-up survey with patients or their proxies was conducted six months later, preferably via telephone or alternatively postal questionnaires.

Data

For the multivariable analysis of factors associated with six-month HRQOL, we considered patient and health care characteristics that have been under investigation in previous studies. In detail, we assumed that the outcomes are influenced by the previous burden of disease and disability, by typical aspects of the geriatric syndrome such as dementia, malnutrition, and mental health. Furthermore, we assumed that complications during the initial hospital stay that led to ICU treatment and the provision of follow-up rehabilitation might have an effect on the recovery of HRQOL. We also sought to examine the role of social factors such as the existence of resources of social support, educational resources, and migration background.

In addition, we adjusted for sex, age, type of hip fracture, type of surgery, study center, the baseline levels of EQ-5D-5L and the baseline hip functionality.

Basic patient characteristics such as sex and age, as well as diagnoses including all diagnosed comorbidities, surgery procedures, and hospital processes such as ICU care or discharge destination were extracted from hospital information systems and medical files. We used self-designed questionnaire items to assess fracture etiology, history of falls, and patients’ living situation.

We assessed the burden of comorbidities by reviewing all documented diagnoses according to the Charlson Comorbidity Index (CCI) (Charlson et al., 1987), a well-established index weighting and classifying the number and seriousness of comorbidities. We used the unadjusted index version of the CCI, and four categories. As a second indicator for the general disease severity and its consequences, we analyzed the number of medications and categorized five and more drug substances as polypharmacy.

Pre-fracture care dependency was determined according to the German long-term care insurance. People with a dependency in their activities of daily living (ADL) are entitled to financial benefits. After a thorough assessment by medical officials, applicants are assigned to a category that describes their degree of dependency and according benefits (Schnitzer et al., 2017). Up until 2016, there were three basic levels, after a law reform five degrees of care dependency were defined—each with higher values representing higher need for support and higher benefits. This transition coincided with the study period, so both systems had to be applied and set into relation. We used the self-reported care dependency status and crosschecked with eCRF data.

Pre-fracture hip functioning was assessed by using the Oxford Hip Score (OHS), a self-report questionnaire developed for patients undergoing total hip replacement (Murray et al., 2007). Its validity and reliability have been confirmed, including for the German version (Naal et al., 2009). The instrument consists of 12 items with responses on a five level Likert scale. The sum score ranges from 0 to 48 (best functionality). It has been recommended as a disease-specific outcome measure in hip fracture patients (Hutchings, Fox & Chesser, 2011).

Signs of malnutrition were screened by using the Short Nutritional Assessment Questionnaire (SNAQ) (Kruizenga et al., 2005). Symptoms of depression and anxiety were assessed by the short screener from the Patient Health Questionnaire (PHQ), the PHQ-4 (Kroenke et al., 2009). The score ranges from 0 to 12 (severe), and moderate or severe symptoms are assumed from a value of 6.

For social support, we adopted one item from the Oslo-3-Items-Social-Support Scale assessing the number of people one can rely on (Dalgard, 1996), and a 3-items-questionnaire from the German Ageing Survey (DEAS) assessing need for more help and support (Engstler & Schmiade, 2013).

For educational and vocational attainment, we used standard questionnaire categories and the CASMIN Educational Classification (Comparative Analysis of Social Mobility in Industrial Nations) to classify responses (Brauns, Scherer & Steinmann, 2003). We stratified educational attainment into: basic (1a –1c: primary and low secondary), intermediate (2a –2c: intermediate and high secondary) and high (3a and 3b: tertiary education).

A basic set of indicators was used to record migration background (Schenk et al., 2006). In accordance with the definition of the Federal Statistical Office of Germany, a migration background was assumed when the participant or at least one parent was born outside Germany. People from former German territories in Eastern Europe who had to leave their home during or after World War II were also included in this group although being born “German” since they had experienced an episode of migration.

The study center variable was dichotomized into university vs. general hospital due to confidentiality reasons and due to uneven distributions of case numbers.

Outcome measures

Health-related quality of life was determined by the generic and widely used EQ-5D questionnaire which has been recommended for studies examining hip fracture outcomes (Parsons et al., 2014). It consists of two components: the first one indicates the health states in five dimensions, including mobility, self-care, usual activities, pain, and anxiety & depression. For each dimension, participants are asked to rank their personal level. The EQ-5D-5L extends its previous version from three to five possible levels ranging from “no problems” (1) to “extreme problems/unable to” (5) (Herdman et al., 2011). Answers to the five dimensions form an individual health profile that can be assigned a summary index score based on societal preference weights for the health state (EuroQol, 2019). The German value set for the EQ-5D-5L was used to calculate the EQ-5D Index value scoring from −0.661 to 1, with values below zero indicating a health state worse than death and 1 indicating full health (Ludwig, von der Schulenburg & Greiner, 2018).

The second component of the EQ-5D is the EQ Visual Analogue Scale (VAS) recording participants’ self-rated health on a scale from 0 (“worst imaginable health”) to 100 (“best imaginable health state”). At baseline, items were modified to assess patients’ unimpaired HRQOL: participants were asked to rate their state of health during the four weeks before the fracture. For non-German speaking participants, the English, Turkish, and Arabic version of the EQ-5D-5L, and for proxy interviews, the EQ-5D-5L proxy version was used.

Cases of death between baseline and follow-up were ascertained by inquiries with the local residents’ registration offices, which hold complete records of all decedents.

Data analysis

Descriptive analyses were performed for patient characteristics and outcome measures, stratified for women and men. For the outcome measures, mean differences between baseline and follow-up values were calculated, both for the individual dimensions of the EQ-5D-5L, the Index value, and the EQ-VAS scores.

For EQ-5D index and EQ-VAS scores multivariable linear regressions were performed. The dependent variables were the follow-up values of these scores. The independent variables were: Baseline value of the respective EQ-5D score, sex, age, proxy-interview (cognitive impairment), educational status, migration background, living situation, comorbidities (CCI), polypharmacy, pre-fracture dependency, pre-fracture hip functionality (OHS), malnutrition, symptoms of depression and anxiety (PHQ-4), social support, subjective need for more support, type of fracture, type of surgery, ICU episode, referral to a rehabilitation facility. Data comes from six different sites, two university and four non-university hospitals. As the number of patients from two sites is very low (n = 4), a mixed model with site as random effect was not applicable. Therefore, the dichotomous variable “study center” was added to account for the data structure.

Regarding the model assumptions, the Variance Inflation Factor and the distributions of the independent variables were analyzed.

For these analyses a multiple imputation was performed. The imputation model includes all independent variables of the analysis model as well as baseline and follow-up value of the EQ-5D index and EQ-VAS scores. The multiple imputation resulted in 10 imputed datasets and the number of iterations was restricted to 20. Results were pooled using Rubin’s rules. Standard errors are reported for estimates and 95% confidence intervals for r-squared values.

The analyses included all surviving patients until follow-up, including living patients that only participated in the baseline interview.

Sensitivity analyses were done for the subset of patients being 65 years old and older, and for a cohort including only patients who participated in the follow-up interview.

All multivariable analyses were done using R version 4.1.2, for imputation the mice package was used (van Buuren & Groothuis-Oudshoorn, 2011).

Due to the exploratory character of the models, we did not apply a certain p-value level to assume variables to be worth considering as relevant for HRQOL at six months after a hip fracture.

Results

Within 25 months of recruiting, 510 eligible patients were approached in the six study centers of which 120 patients declined to participate. Forty-six patients had a legal guardian who could not be reached or declined. Three hundred forty-four patients were enrolled, the majority (97.7%) in four study centers, including two university and two general hospitals. After systematic data verification, 18 cases had to be excluded due to violation of the inclusion criteria. The final cohort consisted of 326 patients with hip fractures.

At six months, 219 participants could be included in the follow-up interview, 68 participants could not be reached or refused the interview. Figure 1 shows the recruitment process in detail, an analysis of the recruitment process and data quality, including response rates and reasons for non-participation, has been published elsewhere (Krobisch et al., 2020).

Figure 1 Flowchart.

Baseline characteristics

Patient and clinical characteristics, stratified for women and men, are displayed in Table 1. The majority of patients were women (67%), mean age of the cohort at time of ED admission was 75.80 years (SD: 12.16), with women (77.8) being six years older than men (71.7). A total of 45.4% had a femoral neck fracture (ICD-10 S72.0), 46.9% a trochanteric fracture (ICD-10 S72.1), and 6.1% a subtrochanteric fracture (ICD-10 S72.2). Five cases, initially diagnosed as usual hip fractures, eventually turned out to be periprosthetic hip fractures.

Table 1 Baseline characteristics of the EMAAge cohort, stratified for women and men.

	All patients n (%)	Women n (%)	Men n (%)	
	326	219 (67.2)	107 (32.8)	
Patient characteristics				
Age, mean (SD)	75.80 (SD 12.16)	77.79 (SD 10.79)	71.73 (SD 13.75)	
Proxy-Interview (cognitive impairment)	54 (16.6)	40 (18.3)	14 (13.1)	
Education				
Basic	149 (45.7)	102 (49.8)	47 (45.6)	
Intermediate	95 (29.1)	70 (34.1)	25 (24.3)	
High	64 (19.6)	33 (16.1)	31 (30.1)	
Migration background	41 (12.6)	26 (12.6)	15 (14.4)	
Living situation				
Independent with others	111 (34.0)	65 (30.1)	46 (43.8)	
Independent alone	156 (47.9)	111 (51.4)	45 (42.9)	
In a facility	54 (16.6)	40 (18.5)	14 (13.3)	
Pre-fracture health state & risk factors				
Comorbidities (CCI)				
0	93 (28.5)	58 (26.5)	35 (32.7)	
1	71 (21.8)	53 (24.2)	18 (16.8)	
2	57 (17.5)	44 (20.1)	13 (12.1)	
≥ 3	105 (32.2)	64 (29.2)	41 (38.3)	
Polypharmacy (≥ 5 medications)	172 (52.8)	121 (55.3)	51 (47.7)	
Pre-fracture dependency	118 (36.2)	92 (42.8)	26 (26.0)	
Pre-fracture hip functionality (OHS: 0-48),
Median (IQR)1	45 (40, 48)	44 (40, 48)	46 (42, 48)	
Malnutrition	74 (22.7)	53 (26.1)	21 (21.2)	
Symptoms of depression & anxiety (PHQ-4) (at least moderate symptoms)1	36 (14.2)	25 (15.2)	11 (12.5)	
Social support: persons to rely on				
None	14 (4.3)	6 (2.9)	8 (7.7)	
1 or 2	111 (34.0)	77 (37.4)	34 (32.7)	
3 to 5	114 (35.0)	82 (39.8)	32 (30.8)	
More than 5	68 (20.9)	39 (18.9)	29 (27.9)	
Subjective need for more support1	119 (46.9)	85 (50.00)	34 (40.5)	
Falls in the past				
Never	79 (24.2)	49 (22.5)	30 (28.6)	
Once in the past 6 months	45 (13.8)	35 (16.1)	10 (9.5)	
More than once in the past 6 months	52 (16.0)	37 (17.0)	15 (14.3)	
ED visits before (last 6 months)	76 (23.3)	49 (23.6)	27 (27.0)	
Fracture and hospital care				
Type of fracture				
Intracapsular (femoral neck)	148 (45.4)	97 (44.3)	51 (47.7)	
Extracapsular: Pertrochanteric	153 (46.9)	104 (47.5)	49 (45.8)	
Extracapsular: Subtrochanteric	20 (6.1)	5 (4.7)	15 (6.8)	
Periprosthetic	5 (1.5)	2 (1.9)	3 (1.4)	
Type of surgery				
Arthroplasty	116 (35.6)	80 (36.5)	36 (33.6)	
Internal fixation	204 (62.6)	133 (60.7)	71 (66.4)	
No surgery	6 (1.8)	6 (2.7)	0	
ICU episode	110 (33.7)	72 (32.9)	38 (35.5)	
Referral to a rehabilitation facility	201 (61.7)	142 (67.3) (5.5)	59 (56.2)	
Geriatric rehabilitation	180 (55.2)	130 (59.4)	50 (46.7)	
Other type of facility	21 (6.4)	12 (5.5)	9 (8.4)	
Length of stay, Median (IQR)	10 (8, 12)	10 (8, 12)	9 (7, 12)	
Six-months mortality	39 (12.0)	27 (12.3)	12 (11.2)	
Notes.

1 Not included in proxy questionnaires.

The vast majority was able to participate in the study by themselves, in 16.5% interviews had to be conducted with a proxy due to cognitive impairments. Approximately 13% had a migration background or migration experience. The proportion of participants with a higher education was larger in men than in women (30.1% vs 16.1%, p = 0.013). The majority lived independently in their own apartment or house, mostly alone, with more women living by themselves than men (51.4% vs 42.9%, p = 0.053). Participants had a high burden of comorbidities (49.7% with a CCI>=2) and limitations in ADL (36.2%). Women were more often dependent in their ADL than men (42.8% vs 26.0%). More than one in five showed signs of malnutrition (22.7%), and 14.2% showed at least moderate symptoms of depression and anxiety.

Pre-fracture hip functioning according to the OHS was good (Median: 45 of 48), however, this only applies to self-answering participants, as the OHS instrument is unsuitable for proxy interviews.

One in four participants had experienced another fall before suffering from the hip fracture, and 29.8% had fallen before in the last six months. In addition, almost one in four patients had visited an ED during the six months before the fracture.

The majority received internal fixation (62.6%). One in three patients had an ICU episode during their hospital stay, and 61.7% were transferred directly to a rehabilitation facility afterwards.

During the six months follow-up 40 (12.0%) patients died. There were no substantial differences between men and women regarding mortality.

Health-related quality of life

At baseline, the majority had no problems in the EQ-5D dimensions self-care (56.3%) and anxiety (55.8%). Patients had at least some problems with general pain (45.0%) and mobility (41.8%). Women had greater problems with pain (61.6% at least some problems vs 52.9%) and activities (55.6% at least some problems vs 37.9%). Overall, the mean EQ-5D index according to the German tariff was 0.70 (SD: 0.32). Women had a lower index value than men (0.66 vs 0.78). The mean self-rated health on the EQ-VAS was 69.9 (SD: 22.9), with no difference between women and men.

At six months, all EQ-5D health dimensions were lower than before the fracture. The largest losses were observed in activities (mean difference: −0.70), mobility (mean difference: −0.55), and self-care (mean difference: −0.42) (see Files 1–5). The mean difference regarding the EQ-5D index was −0.09 (SD: 0.31) points (mean EQ-5D index at six months: 0.63, SD: 0.32) (see Fig. 2). Both women and men rated their health 11 points (SD: 25.13) below their baseline level (mean EQ VAS at six months: 59.38, SD: 22.93) (see Fig. 3). Almost two in three patients reported a worse health state compared to their pre-fracture situation.

Figure 2 Boxplot development of EQ5D index.

Boxplots showing the difference between EQ5D5L index value at baseline and follow-up for both women and men.

Figure 3 Boxplot development of EQ VAS.

Boxplots showing the difference between EQ VAS at baseline and follow-up for both women and men.

In the regression model for the six-month EQ-5D index, three factors appeared to be influential: depressive and anxiety symptoms (β =−0.19, p = 0.001) and a pre-fracture dependency in ADL (β =-0-17, p =0.011). Patients who were transferred to a rehabilitation facility had a better outcome (β =0.08, p =0.051) (see Table 2).

Table 2 Multivariable linear regression model for EQ5D index at 6 months, using multiple imputation.

Predictor	Estimate	SE of regression	Statistic	Degrees of freedom	p-value	
Patient characteristics						
Intercept	0.3514	0.203	1.73	72	0.087	
EQ-5D index baseline value	0.1616	0.092	1.76	23.8	0.092	
Age	0.0009	0.002	0.51	60.2	0.61	
Male sex	−0.0186	0.037	−0.51	86.7	0.613	
General hospital (reference: university hospital)	−0.0572	0.037	−1.54	76.8	0.129	
Education (reference basic)	 	 	 	 	 	
Intermediate	−0.0127	0.041	−0.31	79.8	0.759	
High	0.0145	0.056	0.26	34.3	0.798	
Migration	−0.0474	0.057	−0.83	31.8	0.41	
Living situation (reference independent with others)	 	 	 	 	 	
Independent alone	0.014	0.035	0.4	112.7	0.691	
In a facility	−0.0354	0.079	−0.45	19.5	0.658	
Proxy	0.027	0.08	0.34	82.2	0.736	
Pre-fracture health state & risk factors	 	 	 	 	 	
Comorbidities (CCI) (reference: 0)	 	 	 	 	 	
1	−0.0855	0.054	−1.6	43	0.118	
2	−0.0539	0.058	−0.93	51.1	0.359	
3+	−0.0721	0.058	−1.24	50.8	0.22	
Pre-fracture dependency	−0.1534	0.055	−2.78	30	0.009	
Pre-fracture hip functionality (OHS)	0.0056	0.004	1.53	42.3	0.134	
Malnutrition	0.0165	0.043	0.38	66.7	0.702	
Symptoms of depression & anxiety (PHQ-4)	−0.1993	0.063	−3.15	30.2	0.004	
Social support: persons to rely on	 	 	 	 	 	
3 to 5	0.0406	0.042	0.96	46.2	0.342	
More than 5	0.0082	0.05	0.16	52.4	0.871	
Subjective need	−0.026	0.039	−0.67	67.4	0.503	
Polypharmacy	−0.0609	0.045	−1.35	53	0.184	
Fracture and hospital care	 	 	 	 	 	
Type of fracture (reference: intracapsular)	 	 	 	 	 	
Extracapsular	0.0164	0.06	0.27	53.2	0.785	
Type of surgery (reference: internal fixation)	 	 	 	 	 	
Arthroplasty	0.0812	0.053	1.54	140.7	0.125	
ICU episode	−0.08	0.049	−1.62	23.5	0.119	
Referral to a rehabilitation facility	0.0749	0.041	1.83	78.4	0.07	
n = 278
R-squared: 0.525. CI [0.419; 0.619]
Adjusted R-squared: 0.478. CI [0.367; 0.579]						

For the six-month EQ VAS model, age was removed as an independent variable. The linear regression model including age would have suggested an artificial association indicating higher EQ VAS values with increasing age (see Table 6). Graphical analysis of the association between age and six-month EQ VAS revealed that there was no linear or quadratic association between these two variables. Apart from the intercept, the estimates of the other independent variables did not change considerably with removing age from the model. In the final model, EQ VAS at six months appeared to be positively associated by better pre-fracture hip functioning (β =0.69, p =0.045), a direct transferal to a rehabilitation facility (β =8.05, p = 0.01), and treatment in a university hospital (parameter for general hospital: β =−6.3, p = 0.034). Negative impact was seen for migration background (β =−9.25, p = 0.014), polypharmacy (β =−6.48, p = 0.083), symptoms of depression and anxiety (β =−9.89, p =0.068), and living in an institution (−8.47, p = 0.104) (see Table 3). Our sensitivity analyses including only patients older than 64 years, and only patients with a complete follow-up interview showed no serious divergences in the model parameters (see Tables 7 & 8).

Table 3 Multivariable linear regression model for EQ5D VAS at 6 months, using multiple imputation.

Predictor	Estimate	SE of regression	Statistic	Degrees of freedom	p-value	
Patient characteristics						
Intercept	33.42	16.3	2.06	26.1	0.05	
EQ-VAS baseline value	0.11	0.1	1.3	38.4	0.2	
Male sex	−0.3	3	−0.1	66.7	0.922	
General hospital	−6.3	2.9	−2.15	98.6	0.034	
Education (reference basic)	 	 	 	 	 	
Intermediate	−6.22	3.7	−1.7	39.8	0.098	
High	−1.57	4.2	−0.38	52.6	0.707	
Migration	−9.52	3.8	−2.5	111.2	0.014	
Living situation (reference independent with others)	 	 	 	 	 	
Independent alone	−3.66	2.9	−1.25	95.9	0.213	
In a facility	−8.47	5.1	−1.66	44.8	0.104	
Proxy	−2.47	7.1	−0.35	42.9	0.73	
Pre-fracture health state & risk factors	 	 	 	 	 	
Comorbidities (CCI) (reference: 0)	 	 	 	 	 	
1	−4.13	4.2	−0.98	50.5	0.334	
2	−4.79	4.7	−1.03	57.3	0.308	
3+	−4.93	4.6	−1.08	60.2	0.285	
Pre-fracture dependency	−0.08	4.5	−0.02	29.8	0.985	
Pre-fracture hip functionality (OHS)	0.69	0.3	2.13	21.2	0.045	
Malnutrition	−0.86	3.4	−0.25	76	0.8	
Symptoms of depression & anxiety (PHQ-4)	−9.89	5.2	−1.9	28.9	0.068	
Social support: persons to rely on	 	 	 	 	 	
3 to 5	4.91	3.2	1.56	77.3	0.124	
More than 5	−0.52	3.6	−0.14	120.8	0.886	
Subjective need	−1.65	3.2	−0.52	57.4	0.605	
Polypharmacy	−6.48	3.6	−1.78	44.5	0.083	
Fracture and hospital care	 	 	 	 	 	
Type of fracture (reference: intracapsular)	 	 	 	 	 	
Extracapsular	2.36	4.6	0.52	75.4	0.608	
Type of surgery (reference: internal fixation)	 	 	 	 	 	
Arthroplasty	2.48	4.3	0.58	122.9	0.565	
ICU episode	0.77	3.2	0.24	55.2	0.814	
Referral to a rehabilitation facility	8.05	3.1	2.63	104.8	0.01	
n = 278
R-squared: 0.393, CI [0.296; 0.486]
Adjusted R-squared: 0.335, CI [0.238; 0.432]						

Discussion

The EMAAge study aimed to analyze the development of health-related quality of life during six months after a hip fracture and sought to identify risk factors and modifiable factors. Both the EQ-5D index value and the EQ-VAS decreased steeply. HRQOL outcomes were associated with several patient characteristics and care factors.

Our results confirm the substantial negative impact of hip fractures on HRQOL that has been observed in comparable studies for the six-month period and beyond (Amarilla-Donoso et al., 2020b; Hack et al., 2019; van de Ree et al., 2019).

Regarding the EQ5D index, there were three factors with a noticeable association in our regression model. Patients with an already existing dependency in their ADL at the time of the hip fracture, and patients who reported symptoms of depression and anxiety during the weeks before the fracture, reached worse outcomes. This is in line with the results of several studies that identified the negative impact of depression and pre-fracture ADL limitations (Hack et al., 2019; Kelly-Pettersson et al., 2020).

The model for the self-rated health (EQ-VAS) showed a more complex picture: the outcome was remarkably associated with the sociodemographic variables age and migration background, with pre-fracture limitations in independence (living in a nursing home, hip functionality), pre-fracture health problems (polypharmacy and mental health), and health care characteristics (university hospital vs general hospital, and discharge to a rehabilitation facility). EQ-VAS outcomes have been reported less frequently before. However, the self-rated health state seems to be more prone to patient characteristics and modifiable factors than the health index. For some of the associated variables, sociocultural and psychological mechanisms appear plausible. For instance, the (temporary) loss of mobility might affect patients with better pre-fracture hip functioning more seriously, as they are less prepared for health-related limitations. It has been shown in other health areas that patients with a migration background and/or migration experience are more vulnerable to worse health outcomes due to language barriers, reduced health literacy, an ethnocentric bias in service delivery structures, or a higher burden of stress in the wake of migrating (Rechel et al., 2013; Ward, Kristiansen & Sørensen, 2019).

Interestingly, the burden of comorbidities measured by the CCI was neither associated with the EQ-5D index nor the EQ-VAS. Instead, patients with polypharmacy had considerably lower results for both outcome components. This leaves room for interpretation: polypharmacy could be a surrogate for another dimension of morbidity than measured by the CCI; it could be associated with additional side effects; it could be an obstacle for effective additional medication during rehabilitation; or it might be associated with a different, more negative self-perception of patients. Polypharmacy has been linked to increased risk for falls and subsequent hip fractures (Sergi et al., 2011), as well as readmissions due to new falls (Härstedt et al., 2016). The correlations of patient-centered outcomes and medication have been considered only in few studies. In a rehabilitation-based study, fall-risk increasing drugs were associated with worse functional outcomes (Semel et al., 2010). However, the role of polypharmacy needs to be subjected to further investigations.

At baseline, women reported significantly more health problems according to the five EQ-5D-5L dimensions. This might be explained by women’s higher age in our sample but also by the fact that women seem to be more likely to have more health problems which has been observed in the general population (König et al., 2010). However, gender was not associated with the EQ5D index at six months nor self-rated health (EQ-VAS). While some studies saw gender differences in HRQOL after hip fracture, the majority of comparable studies found no significant effect of gender (Amarilla-Donoso et al., 2020a; Hack et al., 2019; Peeters et al., 2016).

There was no effect of the type of surgery and of ICU episodes during hospital stay but a substantial association was seen between worse outcomes and no (direct) referral to a rehabilitation facility. While there is no sufficient evidence on the most effective rehabilitation programs after hip fracture (Crotty et al., 2010; Sheehan et al., 2019), our results suggest that patients benefit substantially from participation in structured rehabilitation programs after discharge from acute hospital. The importance of rehabilitation and continuity of care after treatment for hip fracture has also been emphasized elsewhere (Pinto et al., 2022).

None of the participating study centers provided any kind of ortho-geriatric care during the enrollment period. This approach comprises different concepts and is increasingly recommended and implemented (Lems et al., 2017). There is sufficient evidence that ortho-geriatric care for hip fracture patients reduces mortality (Rapp et al., 2020), time to surgery (Grigoryan, Javedan & Rudolph, 2014), and improves functional outcomes (Prestmo et al., 2016). HRQOL has not been systematically included as an outcome in studies evaluating ortho-geriatric care. In a Norwegian randomized controlled trial, however, four-months HRQOL (EQ-5D Index) was significantly higher in patients receiving comprehensive geriatric care than in patients receiving standard orthopedic care (Prestmo et al., 2015). Since the beginning of 2021, a new guideline of the German Federal Joint Committee (G-BA) has been in force, which sets new treatment standards for hip fracture patients including early ortho-geriatric care. Further research needs to investigate the potential of these new care models to address some of the identified risk factors for worse outcomes, such as depressive symptoms and the need for structured rehabilitation.

Strengths and limitations

To our knowledge, this is the first multicenter study on quality of life in hip fracture patients in Germany. One of its strengths lies in the study’s inclusiveness. As a patient-centered study in health services research, we sought to capture real-life data of as many affected patients as possible. Therefore, ability to consent and actively participate was not a prerequisite, and both community dwelling and nursing home residents were included. Further, we tried to include patients with migration background by providing questionnaires in the most prevalent immigrant languages in this area. However, the use of questionnaires in foreign languages remained limited. We did not set age limits to include both very old patients and younger adults who can be affected by osteoporotic fractures as well. Another strength is the use of the EQ-5D-5L version which holds better discriminatory properties than the widely used EQ-5D-3L version (Buchholz et al., 2018). We also included important dimensions such as migration background, malnutrition, or polypharmacy in our study that have hardly been investigated in hip fracture studies before. Nevertheless, the list of analyzed factors does not claim to be exhaustive.

The inclusive real-life approach, however, is accompanied by limitations: Our sample is comparatively heterogeneous which can be criticized. Another limitation lies in the fact that a considerable number of participants was lost to follow-up. Due to data protection rules, participants who had moved during the six months after enrollment without letting us know were unable to reach. Among those, there might be relevant number of institutionalized people. Previous studies have reported institutionalization rates of 10–15% within six months after hip fracture (Dyer et al., 2016; Rapp et al., 2015). Clinical data were only available for the initial hospital stay. Therefore, information on the follow-up health care remains limited. This includes the exact volume and type of rehabilitation care. Lastly, our results cannot be easily transferred to other populations. They originate from an inner-city district of a metropolitan city in Germany with specific living and health care conditions.

Conclusions

In our study, we found different risk factors for an above-average deterioration of HRQOL in patients at six months after suffering from a hip fracture. Patients with a particular vulnerability seem to be those with a migration background, nursing home residents, and patients who depend on many drugs. These patient groups need to receive special attention in clinical settings and in research investigating care concepts. Subsequent rehabilitation programs seem to make a major difference regarding HRQOL. The goal should be to reduce the share of patients without rehabilitation, for example by motivating hesitant patients. Furthermore, patients with symptoms of depression and anxiety need adequate care of both consequences of the hip fracture and their mental health problems.

The prevention of hip fractures remains a crucial area of both research and clinical care. One important aspect is the timely identification of patients at risk and initiation of prevention programs. Next to other locations, the ED might hold the potential to improve prevention due to its central position within the complex health system. As we have shown, a considerable number of patients experienced milder falls and was seen in an ED before.

Supplemental Information

Supplemental Information 1 R code for the preparation of data for multivariable analyses

Click here for additional data file.

Supplemental Information 2 R code regression models including imputation

R code for linear regression models for EQ5D index and EQ VAS at 6 months including multiple imputation and the code for the model for EQ VAS including the variable age.

Click here for additional data file.

Supplemental Information 3 R code for the sensitivity analyses age 65+ for linear regression models for EQ5D index and EQ VAS at 6 months including multiple imputation

Click here for additional data file.

Supplemental Information 4 R code for the sensitivity analyses without dropouts for linear regression models for EQ5D index and EQ VAS at 6 months including multiple imputation

Click here for additional data file.

Supplemental Information 5 Boxplot Development EQ5D Mobility

Boxplot showing the difference between EQ5D Mobility dimension at baseline and follow-up - both for women and men

Click here for additional data file.

Supplemental Information 6 Boxplot Development EQ5D Selfcare

Boxplot showing the difference between EQ5D Selfcare dimension at baseline and follow-up - both for women and men

Click here for additional data file.

Supplemental Information 7 Boxplot Development EQ5D Activity

Boxplot showing the difference between EQ5D Activity dimension at baseline and follow-up - both for women and men

Click here for additional data file.

Supplemental Information 8 Boxplot Development EQ5D Anxiety

Boxplot showing the difference between EQ5D Anxiety dimension at baseline and follow-up - both for women and men

Click here for additional data file.

Supplemental Information 9 Boxplot Development EQ5D Pain

Boxplot showing the difference between EQ5D Pain dimension at baseline and follow-up - both for women and men

Click here for additional data file.

Supplemental Information 10 Sensitivity analysis for EQ VAS - model including variable age

Sensitivity analysis for the regression model for EQ VAS (Table 3), model including the variable age which was removed in the main model due to non-linear associations between age and the outcome and artificial results

Click here for additional data file.

Supplemental Information 11 Sensitivity analysis for EQ VAS - model including only age 65+

Sensitivity analysis for the regression model for EQ VAS (Table 3), model including only subjects aged 65 and older

Click here for additional data file.

Supplemental Information 12 Sensitivity analysis for EQ VAS - model including no dropout cases

Sensitivity analysis for the regression model for EQ VAS (Table 3), model including only subjects with complete follow-up data (dropout cases excluded)

Click here for additional data file.

The authors would like to thank the participating hospitals in the EMANet research network, namely Charité –Universitätsmedizin Berlin (Campus Charité Mitte and Campus Virchow-Klinikum), St. Hedwig Hospital (Alexianer St. Hedwig-Krankenhaus Berlin), Elisabeth Hospital (Evangelische Elisabeth Klinik der Paul-Gerhardt Diakonie), Franziskus Hospital (Franziskus-Krankenhaus Berlin), German Armed Forces Hospital Berlin (Bundeswehr Krankenhaus Berlin), German Red Cross Hospital Berlin-Mitte (DRK Kliniken Berlin-Mitte), and the Berlin Jewish Hospital (Jüdisches Krankenhaus Berlin). The authors would like to thank all study participants as well as all EMANet researchers and study personnel responsible for study planning, patient recruitment, and data management.

Abbreviations

ADL Activities of Daily Living

CASMIN Comparative Analysis of Social Mobility in Industrial Nations

CI Confidence Interval

CCI Charlson Comorbidity Index

DALY Disability-adjusted Life Years

DEAS German Ageing Survey (Deutscher Alterssurvey)

eCRF Electronic Clinical Report Form

ED Emergency Department

EMANet Emergency and Acute Medicine Network for Health Care Research

EQ VAS EuroQol Visual Analog Scale

EQ-5D-5L EuroQol-5D-5L questionnaire

HRQOL Health-Related Quality of Life

ICD-10 10th Revision of the International Classification of Diseases

ICU Intensive Care Unit

IQR Interquartile Range

OHS Oxford Hip Score

PHQ-4 Patient Health Questionnaire-4

SD Standard Deviation

SNAQ Short Nutritional Assessment Questionnaire

Additional Information and Declarations

Competing Interests

Author Contributions

Human Ethics

Data Deposition

The authors declare there are no competing interests.

Johannes Deutschbein conceived and designed the study, performed the data collection, analyzed the data, prepared figures and/or tables, authored drafts of the article, and approved the final draft.

Tobias Lindner conceived and designed the study, authored or reviewed drafts of the article, and approved the final draft.

Martin Möckel conceived and designed the study, authored or reviewed drafts of the article, initiated the research network EMANet, he also is principal investigator and speaker of the network, and approved the final draft.

Mareen Pigorsch analyzed the data, prepared figures and/or tables, authored or reviewed drafts of the article, and approved the final draft.

Gabriela Gilles analyzed the data, prepared figures and/or tables, authored or reviewed drafts of the article, and approved the final draft.

Ulrich Stöckle conceived and designed the study, authored or reviewed drafts of the article, and approved the final draft.

Ursula Müller-Werdan conceived and designed the study, authored or reviewed drafts of the article, and approved the final draft.

Liane Schenk conceived and designed the study, authored or reviewed drafts of the article, co-speaker of the research network EMANet, and approved the final draft.

The following information was supplied relating to ethical approvals (i.e., approving body and any reference numbers):

The ethics committee of Charité—Universitätsmedizin Berlin approved of this study.

The following information was supplied regarding data availability:

The raw measurements and the code for multivariable analyses were made available for peer review. The data cannot be shared publicly due to the lack of clear patient consent.

It will be available at Zenodo after users have committed to a data sharing agreement which restricts data usage to scientific purposes and prohibits distributing the data in order to protect confidentiality.

Please contact the data manager, Elke Matheis at medsoz@charite.de, for access to the dataset: 10.5281/zenodo.7534926.

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
