# Peer review of "Health-related quality of life and associated factors after hip fracture. Results from a six-month prospective cohort study"

_PeerJ, doi:10.7717/peerj.14671_

## Round 0.1 · original submission · Major Revisions

Dear Authors, your manuscript has been assessed by the relevant experts and raised some concerns about the power, methods, and presentation of the findings. You are requested to consider the comments and submit your revised manuscript at your earliest.

Reviewer 1 ·

Basic reporting

Dear Authors

After reviewing your manuscript I have some questions to raise.

First of all, after reviewing the study protocol registered in the German Clinical Trials Register (DRKS00014273), I have noticed that of the sample initially indicated in the protocol (350) you start the study with 326 participants and end it with 219.

I have not been able to find in your article any concrete data explaining how this situation affects the power and effect size studied in your work.
Please, could you include that information?

Experimental design

The authors describe with sufficient details to the research question is well defined and the methodology allows them to replicate their study.

Validity of the findings

The data provided are robust and the analyses are statistically adequate.

The limitations of their study are well defined.
Conclusions are linked to original research questions.

Additional comments

No comments.

·

Basic reporting

Hip fracture is a common disease in the elderly, which seriously affects the life span and quality of life of the elderly. In this study, the researchers found the factors affecting the quality of life of postoperative hip fracture in the elderly. This study is a prospective cohort study with good clinical value and innovation. The article has clear logic and accurate language expression.

Experimental design

1. In this study, the researchers eventually recruited 326 subjects. Please authors respond: how is the sample size calculated?
2. The author is asked to explain the descriptive statistical methods, statistical test methods and test standards in data Analysis. In particular, the researchers followed up the patients with hip fracture for 6 months, observed the relevant clinical indicators before and after 6 months, and made a statistical analysis, and asked the author to explain the statistical methods used.

Validity of the findings

Figures 2 and 3 are simple box diagrams, the content of the picture itself can not show the relevant data expressed by the author in the article, please explain.

Additional comments

No.

---

## Round 0.2 · Minor Revisions

Thank you for incorporating the suggestions from the reviewer. I have a few concerns that should be addressed by the authors.

The title of the manuscript should be amended to: "Health-related quality of life after hip fracture; results from a six-month prospective cohort study." It will be more efficient if the title is presented in a way that states the objective of the study, i.e., evaluation of HRQOL........

The analysis is mainly focused on gender; however, the title does not make any such claim. The authors are requested to modify the title as per the objective of the study.

The gender comparison has been done in the results section, but the discussion does not provide any information on such a comparison. It is important to highlight why gender comparison is required for hip fracture patients. Is there any effect of gender on the outcomes of hip fractures among patients? provide a gender-wise comparison with other studies conducted on the same topic at national and international levels.

how the lack of male gender representation in the current study will affect the findings and generalizability of the results

In the methodology section, please include information on the EQ-5D construct, 5-item Likert scale, VAS details, and interpretations. The readers would benefit from the study if they could understand the basic construct of EQ-5D as there are numerous tools for the estimation of QOL.

·

Basic reporting

The manuscript has complete structure, accurate language and fluent expression.

Experimental design

Design is scientific and reasonable.

Validity of the findings

The data are full and accurate, and the results are true and credible.

Additional comments

No

---

## Round 0.3 · accepted · Accept

Dear Authors, thank you for incorporating the suggested changes in the manuscript.